# The Functional Properties and Physiological Roles of Signal-Transducing Adaptor Protein-2 in the Pathogenesis of Inflammatory and Immune Disorders

**DOI:** 10.3390/biomedicines10123079

**Published:** 2022-11-30

**Authors:** Jun-ichi Kashiwakura, Kenji Oritani, Tadashi Matsuda

**Affiliations:** 1Department of Life Science, Faculty of Pharmaceutical Sciences, Hokkaido University of Science, Sapporo 0068585, Hokkaido, Japan; 2Department of Hematology, International University of Health and Welfare, Narita 2868686, Chiba, Japan; 3Department of Immunology, Graduate School of Pharmaceutical Sciences, Hokkaido University, Sapporo 0600812, Hokkaido, Japan

**Keywords:** signal-transducing adaptor protein-2 (STAP-2), signal transduction, immune response, T cells, T cell antigen receptor (TCR)

## Abstract

Adaptor molecules play a crucial role in signal transduction in immune cells. Several adaptor molecules, such as the linker for the activation of T cells (LAT) and SH2 domain-containing leukocyte protein of 76 kDa (SLP-76), are essential for T cell development and activation following T cell receptor (TCR) aggregation, suggesting that adaptor molecules are good therapeutic targets for T cell-mediated immune disorders, such as autoimmune diseases and allergies. Signal-transducing adaptor protein (STAP)-2 is a member of the STAP family of adaptor proteins. STAP-2 functions as a scaffold for various intracellular proteins, including BRK, signal transducer, and activator of transcription (STAT)3, STAT5, and myeloid differentiation primary response protein (MyD88). In T cells, STAP-2 is involved in stromal cell-derived factor (SDF)-1α-induced migration, integrin-dependent cell adhesion, and Fas-mediated apoptosis. We previously reported the critical function of STAP-2 in TCR-mediated T cell activation and T cell-mediated autoimmune diseases. Here, we review how STAP-2 affects the pathogenesis of T cell-mediated inflammation and immune diseases in order to develop novel STAP-2-targeting therapeutic strategies.

## 1. Introduction

T cells are essential players in acquired immune responses. Antigen-presenting cells, such as dendritic cells and macrophages, phagocytose pathogens and digest them into peptides, which are then presented to the major histocompatibility complex (MHC) for recognition by T cells. The T cell receptor (TCR) aggregates, and the TCR signal cascade is activated. T cell activation after TCR aggregation is initiated by the lymphocyte-specific protein tyrosine kinase (LCK)-mediated phosphorylation of the TCR-associated CD3 complex and ζ chain. Zeta-chain-associated protein kinase 70 (ZAP-70) is recruited to phosphorylated CD3 molecules, resulting in the phosphorylation of the linker for the activation of T cells (LAT) [1,2,3,4,5]. LAT phosphorylation enhances the interaction of the LAT with several TCR signal molecules, such as glutamic acid decarboxylase (Gad), growth factor receptor-bound protein-2 (Grb2), and phospholipase C gamma 1 (PLC-γ1) [6,7]. IL-2-inducible tyrosine kinase (ITK) is also phosphorylated by LCK and recruited to the LAT, following which ITK phosphorylates PLC-γ1 to initiate the downstream signaling events [8,9]. PLC-γ1 phosphorylation is required for membrane-associated phosphatidylinositol 4,5-bisphosphate (PIP2) catalysis, resulting in the production of inositol trisphosphate (IP3) and diacylglycerol (DAG). The production of IP3 increases the intracellular calcium concentration, followed by the activation of the nuclear factor of activated T cells (NFAT) transcriptional activity, resulting in the induction of interleukin (IL)-2 transcription in T cells [10]. Thus, several adaptor molecules are critical for the activation of TCR signaling cascades.

### Contribution of Adaptor Proteins in T Cell Development and TCR-Mediated T Cell Activation

T cells express several adaptor proteins, such as LAT, SLP-76, VAV, and Grb2. Following ZAP-70 phosphorylation by LCK, activated ZAP-70 phosphorylates the LAT through the LCK-mediated bridging of ZAP-70 to the LAT [11]. ZAP-70 phosphorylates SLP-76 and via the activation of the LAT, indirectly recruits SLP-76 to the LAT in lipid rafts by the binding of Gads. The activation through SLP-76 also affects T cell development and activation. This stepwise activation is crucial for initiating downstream TCR signal transduction using mice or cells that are deficient in these molecules. For example, LAT deficiency results in double-positive thymocyte development failure because the TCR-mediated activation signal is lost at the DN3 stage [12]. LAT-deficient Jurkat T cells are unresponsive to TCR stimulation [13,14]. Mature T cells are absent in SLP-76-deficient mice because thymocyte development in SLP-76-deficient mice does not proceed beyond the DN3 stage [15]. Similarly, SLP-76-deficient Jurkat cells showed no TCR-mediated activation events, such as the phosphorylation of PLC-γ1 and MAPK or IL-2 transcription [16]. The Gab family of proteins functions as scaffold proteins in TCR signaling cascades. Gab2, a member of the Gab family of adaptor proteins, does not affect lymphocyte maturation because the normal number of mature lymphocytes in spleens is shown in Gab2-deficient mice [17]. Gab2 is phosphorylated by ZAP-70 and acts as an inhibitory adaptor protein by recruiting SHP-2 after TCR aggregation [18]. Gab2 associates with the LAT after TCR aggregation, and its PXXXR motif within the MBD domain is crucial for the constitutive association of Gab2 with Gads/Grb2. This constitutive association is required for the recruitment of Gab2 to the lipid rafts after TCR engagement for inducing the inhibitory function of Grab2 in activated T cells [19].

## 2. Signal-Transducing Adaptor Protein Family of Proteins

The signal-transducing adaptor protein (STAP) family of proteins consists of two members: STAP-1 and STAP-2 [20]. Both STAP-1 and STAP-2 contain a pleckstrin homology (PH) domain and an Src homology 2 (SH2) domain in their N- and C-terminal (or the central region in case of STAP-2) regions, respectively [20]. The STAP-2 C-terminal region also has a proline-rich region and YXXQ motif [21]. Human STAP-1 was originally identified as a TEC docking protein [22]. Mouse STAP-1 was cloned as a c-Kit binding protein as well as a signal transducer and activator of transcription (STAT)5 interacting protein [23]. Subsequently, three isoforms of STAP-1 that are dependent on alternative splicing were reported. The authors demonstrated that only full-length STAP-1 activates cAMP-response element binding protein (CREB) activity after B cell receptor (BCR) aggregation in Ramos cells [24]. We previously reported that STAP-1 is a key molecule that regulates chronic myeloid leukemia (CML) stem cell survival by increasing the anti-apoptotic gene expression via enhanced STAT5 activity. We also indicated a direct interaction between STAP-1, STAT5, and BCR–ABL in CML cells [25,26].

Human STAP-2 was originally identified as BKS, i.e., a substrate of BRK tyrosine kinase [27]. We identified murine STAP-2 as a c-fms-interacting protein and revealed that murine STAP-2 modulates STAT3 activity by functioning as an adaptor protein after leukemia inhibitory factor (LIF) or epidermal growth factor (EGF) stimulation [21]. Subsequently, we identified STAP-2 interacting molecules, as shown in Figure 1 [20]. For example, STAP-2 interacts with STAT5 through its PH domain after the Brk-mediated phosphorylation of STAP-2 in breast cancer cell lines, and this interaction may be important for the regulation of breast cancer cell growth [28]. The PH domain- or SH2 domain-deleted STAP-2-overexpressing murine pro-B cell line, Ba/F3 cells, showed enhanced cell growth compared with that of STAP-2 FL-overexpressing Ba/F3 cells [29]. STAP-2 is also associated with c-Fms or macrophage colony-stimulating factor (M-CSF) receptor (M-CSFR). The c-Fms/M-CSFR complex is essential for the differentiation and survival of macrophages and osteoclasts. STAP-2 binds to c-Fms/M-CSFR through its PH domain, and this interaction is enhanced after M-CSF stimulation. We also inferred that the association of STAP-2 with c-Fms/M-CSFR is a critical step for the negative regulation of c-Fms/M-CSFR signaling because STAP-2-overexpressing Raw 264.7 cells reduced the phosphorylation of c-Fms/M-CSFR and its downstream molecules, Akt and ERK, resulting in the inhibition of cell migration in STAP-2-overexpressing Raw 264.7 cells compared with that in mock-transfected Raw 264.7 cells [30]. In accordance with these results, cell motility and c-Fms/M-CSFR signaling were enhanced in STAP-2 knockout (KO) macrophages compared with that in wild-type (WT) macrophages [31].

In addition, STAP-2 has four putative tyrosine residues that are phosphorylated by several kinases [21]. The tyrosine 250 residue within STAP-2 is the most important tyrosine residue for its functions. BRK is one of the kinases which phosphorylates STAP-2 tyrosine 250 (Y250). The BRK-mediated phosphorylation of STAP-2 Y250 is essential for the induction of STAT3 and STAT5 activities [28,32]. STAP-2 Y250 phosphorylation was also observed in cells co-transfected with Src and Jak2 tyrosine kinases [33] and in cells stimulated with LIF [33] and M-CSF [30].

## 3. Functional Role of STAP-2 in T Cells

### 3.1. STAP-2 in Non-TCR-Mediated Signal Transduction

T cells are essential for acquired immune responses to eliminate pathogens such as viruses and cancers. We previously reported that STAP-2 is crucial for certain T cell activation events. STAP-2 deficiency results in the reduction of SDF-1α-induced T cell migration, whereas Jurkat cells overexpressing STAP-2 show increased cell migration toward SDF-1α [34]. Mechanical studies have suggested that STAP-2 affects the activity of Rho GTPases, such as RhoA, Rac1, and Cdc42, with its ability to function as a scaffold protein between Vav1 and Pyk2 to initiate Pyk2-mediated Vav1 phosphorylation in T cells [34,35]. STAP-2 Y250 phosphorylation is also an important step for the successful function of an adaptor protein between these molecular types of machinery [35].

Integrin signaling is required for many cellular processes, such as cell adhesion, cell migration, proliferation, differentiation, and metastasis [36]. One of the important kinases involved in signal transduction through integrins is focal adhesion kinase (FAK). FAK is localized to focal adhesion contacts and phosphorylated in response to integrin–extracellular matrix interactions. STAP-2 KO T cells showed increased fibronectin (FN)-mediated cell adhesion compared with that in WT T cells because FAK expression levels were increased with STAP-2 deficiency. STAP-2 directly associates with FAK via its SH2 domain. STAP-2 recruits Cbl, an E3 ubiquitin ligase, to FAK, and this interaction promotes the ubiquitin-dependent degradation of FAK, resulting in the reduction of T cell adhesion [37].

STAP-2 is also involved in T cell survival [38], which is strictly regulated to maintain T cell numbers at physiological levels. Once the dysregulation of T cell survival occurs, autoreactive T cells enter the periphery, and autoimmune diseases, such as systemic lupus erythematosus and Sjörgren’s syndrome, develop. The Fas/Fas ligand interaction is crucial for the induction of apoptosis in activated T cells to eliminate autoreactive T cells. In fact, T cells from Fas-mutated mice (MRL/Mp-lpr/lpr mice) escape cell elimination in the thymus, and these mice show systemic lupus erythematosus-like phenotypes, such as increased anti-DNA antibodies, progressive renal failure, and lymphadenopathy. STAP-2-overexpressing Jurkat cells were more susceptible to FAS-induced apoptosis than mock-transfected Jurkat cells. STAP-2 binds directly to caspase-8, a key caspase family protein that controls apoptosis. The STAP-2/caspase-8 complex is recruited to FADD, resulting in the activation of caspase-3. These data possibly suggest an important role for STAP-2 in FAS-mediated T cell apoptosis.

### 3.2. STAP-2 in TCR-Mediated Signal Transduction

STAP-2, in addition to its contributions to non-TCR-mediated T cell function, also enables TCR-mediated T cell activation [39]. STAP-2 KO T cells impair cell proliferation and IL-2 production after TCR stimulation. STAP-2 KO T cells also showed reduced phosphorylation of TCR signaling molecules, such as ZAP-70 and PLC-γ1. Immunofluorescence and immunoprecipitation studies have demonstrated that STAP-2 associates with LCK in a stimulation-dependent manner. STAP-2 also interacts with CD3ζ, a substrate of LCK. STAP-2-overexpressing Jurkat cells showed higher levels of CD3ζ phosphorylation than the control Jurkat cells, indicating that STAP-2 functions as a scaffold protein between LCK and CD3ζ (Figure 2). STAP-2 Y250 phosphorylation is required for the strong association of STAP-2 with LCK, as evidenced by the decreased association between STAP-2 and LCK in STAP-2 Y250F-overexpressing mutant cells. Thus, STAP-2 is a positive regulator of TCR signaling and may be a good therapeutic target for T cell-mediated immune disorders.

## 4. Pathophysiological Role of STAP-2 in T Cells

### 4.1. STAP-2 in Infection

The Toll-like receptor (TLR) family is an important pattern recognition receptor for the elimination of pathogens, such as bacteria and viruses [40]. In certain TLR signaling pathways, MyD88 functions as an adaptor protein for the production of inflammatory cytokines, such as IL-6 and tumor necrosis factor (TNF)-α, which cause septic shock. MyD88-deficient mice are more resistant to sepsis caused by polymicrobial infection than WT mice [41]. We previously reported that STAP-2 interacts with MyD88 and IKK α/β through its SH2-like domain and is required for the maximal production of inflammatory cytokines in macrophages after lipopolysaccharide (LPS)- or CpG-stimulation [42], indicating that STAP-2 is essential for the induction and development of innate immune responses. Indeed, STAP-2 in T cells is involved in chronic infectious diseases.

*Propionibacterium acnes* (*P. acnes*)-induced granuloma formation is an infection model in which mice show splenomegaly and granuloma formation in the liver in response to heat-killed *P. acnes* in a Th17 response-dependent manner [43,44]. STAP-2 KO mice showed less splenomegaly after the *P. acnes* administration than WT mice. The granuloma area was significantly reduced in STAP-2 KO mice than in WT mice after the *P. acnes* injection. In contrast, lymphocyte-specific STAP-2 transgenic (Tg) mice showed severe splenomegaly and granuloma formation after the *P. acnes* treatment compared with WT mice [39]. Therefore, STAP-2 is likely to be critical for controlling infections.

### 4.2. STAP-2 in Autoimmune Diseases

In autoimmune diseases, the immune system attacks the body’s own organs and destroys healthy tissues. The dysregulation of T cells appears to be a cause of certain autoimmune diseases, such as multiple sclerosis. Using experimental autoimmune encephalomyelitis (EAE), a mouse model of multiple sclerosis, the Th17 response was proposed to be necessary for the pathogenesis of EAE [45]. The Th17 response is also essential for the pathogenesis of psoriasis, and a new therapy based on targeting IL-17 or IL-17 signaling molecules has established a good strategy for treating patients with psoriasis. STAP-2 in T cells contributes to Th17 generation and Th17-mediated autoimmune diseases. STAP-2 deficiency results in the significant suppression of disease severity after EAE induction, whereas lymphocyte-specific STAP-2 Tg mice are more severely affected than the WT mice after EAE induction. Th17 infiltration in the spinal cord tends to be reduced in STAP-2 EAE mice compared with that in WT EAE mice, whereas Th17 infiltration is significantly increased in lymphocyte-specific STAP-2 Tg mice than in WT mice. Therefore, STAP-2 expression may be involved in the development of T cell-dependent autoimmune diseases.

### 4.3. STAP-2 in Tumorigenesis

Understanding the mechanisms underlying leukemia and solid tumors is necessary for the development of new therapeutic strategies for patients. One of the recent innovative therapies is the use of immune checkpoint inhibitors, such as ipilimumab (anti-CTLA-4 mAb) and nivolumab (anti-PD-1 mAb). Although these new therapeutic medicines constitute a significant breakthrough, the response rate of patients with solid tumors is approximately 30%, indicating that other targets need to be identified to cure patients with malignancies. In this regard, STAP-2 may be a good target for treating patients with certain types of leukemia and solid tumors.

BCR–ABL is a chimeric protein derived from the Philadelphia chromosome. More than 90% of patients with CML have Philadelphia chromosome abnormalities and BCR–ABL expression, which causes the onset of CML. STAP-2 binds to BCR–ABL through its SH2-like domain [46]. STAP-2-overexpression results in increased cell survival in Ba/F3 cells by enhancing the phosphorylation of BCR–ABL and STAT5. STAP-2 Y250 is phosphorylated by BCR–ABL and is required for the BCR–ABL-induced enhancement of cell survival. Thus, STAP-2 is a good target for developing therapeutic medicines for patients with leukemia, especially CML.

STAP-2 also participates as a scaffold protein in the EGF receptor (EGFR) signaling cascade [47]. STAP-2 knockdown results in a reduction in human prostate cancer proliferation and tumorigenesis. Mechanistically, STAP-2 regulates the EGF-induced phosphorylation of STAT3, Akt, and ERK through its direct interaction with the EGFR. STAP-2 Y250 phosphorylation induced by EGFR is required for the interaction of STAP-2 with EGFR. Notably, the EGFR/STAP-2 interaction is essential for sustaining EGFR expression on the surface of tumor cells by inhibiting the c-CBL-mediated ubiquitination of EGFR.

STAP-2 is also involved in the growth of breast cancer cells through its interaction with Brk and STAT3 [47]. During immune surveillance by T cells, STAP-2 may promote the generation of long-term memory CD8^+^ T cells [48]. STAP-2 suppresses terminal effector CD8^+^ T cell generation, resulting in the promotion of tumor antigen-reactive memory CD8^+^ T cell development, although the detailed mechanism remains to be elucidated. Thus, STAP-2 is likely to be a suitable target for the development of novel therapeutic drugs for patients with certain types of cancer, specifically leukemia.

## 5. Conclusions

As described here, STAP-2 interacts with a variety of key molecules involved in cell signaling and transcription. The manipulation of STAP-2 expression was shown to contribute to inflammatory and immune diseases and some malignancies critically. Hence, we hope that a novel STAP-2-targeting drug is developed in the future.

## Figures and Tables

**Figure 1 biomedicines-10-03079-f001:**
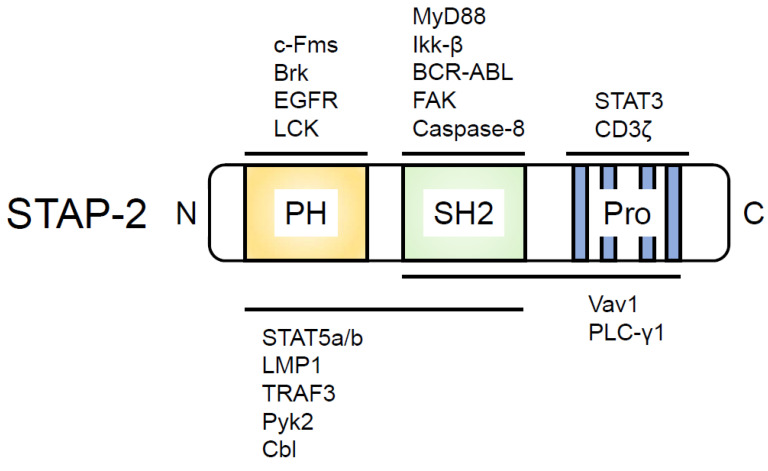
STAP-2 structure and domain-binding proteins. PH, Pleckstrin homology domain; SH2, Src homology 2 (SH2) domain; Pro, Proline-rich region containing a YXXQ motif. The proteins that bind to the domain within STAP-2 are indicated above or below the illustrated STAP-2.

**Figure 2 biomedicines-10-03079-f002:**
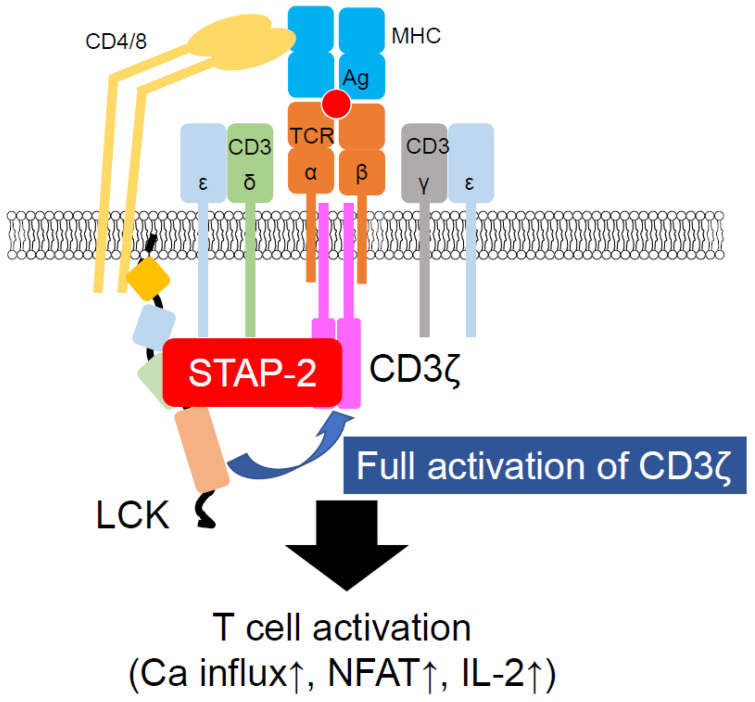
Participation of STAP-2 in LCK-mediated CD3ζ activation. STAP-2 functions as an adaptor protein of LCK and CD3ζ. Once these proteins bind to STAP-2, LCK easily phosphorylates CD3ζ, resulting in the induction of T cell activation events, such as calcium mobilization and IL-2 production.

## Data Availability

Not applicable.

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
