# Peer review of "The Functional Properties and Physiological Roles of Signal-Transducing Adaptor Protein-2 in the Pathogenesis of Inflammatory and Immune Disorders"

_biomedicines, 2022, doi:10.3390/biomedicines10123079_

Round 1
Reviewer 1 Report
The authors review the functions performed by signal-transducing adapter protein-2 (STAP-2) in health situations and in some diseases. In general, the article is correct and adequately condenses the knowledge about STAP-2.
Comments:
The title is inappropriate. And this for several reasons: 1) The article is not only about new ("Novel") contributions, but it is a review of what is known about STAP-2. 2) Much of the work deals with the physiological functions of STAP-2 and not only pathology. 3) A notable chapter is the role exerted on macrophages (and not just T lymphocytes) and their involvement in the synthesis of proinflammatory cytokines. 4) It also deals with the involvement of the protein in neoplasms.
The bibliography is somewhat outdated and does not include several of the recent articles referring to STAP-2 and its involvement in inflammatory and autoimmune diseases
Reviewer 2 Report
The manuscript reviews how adaptor protein STAP-2 affects the pathogenesis of T cell-mediated inflammation, autoimmune diseases and cancer, suggesting novel therapeutic strategies by targeting STAP-2.
The work is concise, informative and appropriate for the aim of the study.
Overall, the manuscript needs no revision and can be published in its present form.
Round 2
Reviewer 1 Report
The article may be published in its current form. The corrections respond to the aspects raised in the previous review